# Supramolecular β-Cyclodextrin-Quercetin Based Metal–Organic Frameworks as an Efficient Antibiofilm and Antifungal Agent

**DOI:** 10.3390/molecules28093667

**Published:** 2023-04-23

**Authors:** Rajaram Rajamohan, Chaitany Jayprakash Raorane, Seong-Cheol Kim, Mani Murali Krishnan, Yong Rok Lee

**Affiliations:** 1School of Chemical Engineering, Yeungnam University, Gyeongsan 38541, Republic of Korea; 2Department of Chemistry, Bannari Amman Institute of Technology, Sathyamangalam 638401, India

**Keywords:** β-cyclodextrin, quercetin, metal–organic framework, biofilm, *Candida albicans* DAY185 strain

## Abstract

The loading of drugs or medicinally active compounds has recently been performed using metal–organic frameworks (MOFs), which are thought to be a new type of porous material in which organic ligands and metal ions can self-assemble to form a network structure. The quercetin (QRC) loading and biofilm application on a cyclodextrin-based metal–organic framework via a solvent diffusion approach is successfully accomplished in the current study. The antibacterial plant flavonoid QRC is loaded onto β-CD-K MOFs to create the composite containing inclusion complexes (ICs) and denoted as QRC:β-CD-K MOFs. The shifting in the chemical shift values of QRC in the MOFs may be the reason for the interaction of QRC with the β-CD-K MOFs. The binding energies and relative contents of MOFs are considerably changed after the formation of QRC:β-CD-K MOFs, suggesting that the interactions took place during the loading of QRC. Confocal laser scanning microscopy (CLSM) showed a reduction in the formation of biofilm. The results of the cell aggregation and hyphal growth are consistent with the antibiofilm activity that is found in the treatment group. Therefore, QRC:β-CD-K MOFs had no effect on the growth of planktonic cells while inhibiting the development of hyphae and biofilm in *C. albicans* DAY185. This study creates new opportunities for supramolecular β-CD-based MOF development for use in biological research and pharmaceutical production.

## 1. Introduction

Metal–organic frameworks (MOFs) have just recently come to light as crystalline porous materials with potential use in biological applications, including imaging and drug administration [1]. MOFs are particularly appealing for the inclusion of significant pharmacological payloads, due to their high porosity, vast surface areas, and diversity in terms of composition and functions [2,3,4]. Using excipients approved by the pharmaceutical industry or endogenous linkers, numerous attempts have been made to manufacture biocompatible MOFs [5,6,7].

In pharmaceutical-related research, particularly food technology, biotechnology, and analytical chemistry, cyclodextrins (CDs) are well known, easily accessible, and often used [8]. These molecules feature a hydrophilic exterior surface and a hydrophobic interior cavity, which can interact non-covalently with a range of biologically active chemicals [9]. A relatively recent family of metal–organic frameworks is CD-MOFs [10]. To create structures with channels and cavities, CD-MOFs are put together from native CDs and metal ions. As a result, CD-MOFs have a large specific surface area, which makes it possible to encapsulate medicinal molecules. The majority of the time, but not always, CD-MOFs have porous structures. According to Liu et al. [11] and Ding et al. [12], the type of metal ion and linker, as well as the synthetic conditions, impact the availability and also the size of pores that help to transform or convert the material of MOFs for drug delivery. Nowadays, MOFs based on CD are more commonly used for drug encapsulation [9,13,14,15]. To the best of our knowledge, there is nothing written about encapsulation in CD-MOFs based on α-CD and β-CD. For instance, Sha et al. [16] synthesized a new α-CD-MOF and studied how it might be used as a drug carrier. A β-CD-MOF containing chiral helices is effectively synthesized and described [17]. Liu et al. [11] investigated the effects of template agents (ibuprofen or methyl benzene sulfonic acid, 1,2,3-triazole-4,5-dicarboxylic acid) on the crystallization of β-CD-MOFs. The loading capacity of CD-MOFs and the kinetics of the drug release have been demonstrated to be significantly impacted by the size and shape of the pores. The use of MOFs in the food sector has recently increased; they are mostly used as antibacterial agents and carriers for food packaging applications [2,18,19,20,21,22]. The food sector prioritizes MOFs with strong biocompatibility and non-toxic properties because of their applications.

A plant flavanol from the polyphenolic flavonoid family is known as quercetin (QRC). It is present in a wide variety of fruits, vegetables, leaves, seeds, and grains; typical foods containing significant levels of QRC include capers, red onions, and kale [23]. QRC has antioxidant, anti-atherogenic, and anti-carcinogenic effects similar to several other bioflavonoids [24,25,26]. Unfortunately, its limited water solubility severely restricts its use in food and medicine. By forming inclusion complexes (ICs) with the native and modified β-CDs, the issue of solubility has been defeated. QRC has recently been tested for a number of significant biological applications, including drug release [27,28], solubility improvement [29,30,31], wound healing [30,31], and in vitro cytotoxic potential [31]. Recently, new porous carriers for loading functional goods, called β-CD-K MOFs, have been made with QRC for cancer therapy [32]. Hence, in order to accomplish converting the biologically significant and potentially innovative material of QRC, CD-based MOFs must be created. Based on the aforementioned data, it is necessary to develop a novel class of MOFs with inclusion-based components by loading QRC for biologically significant anti-fungal activities, as well as for a decreased capacity for biofilm formation.

This study aimed to determine the feasibility of using β-CD-K MOFs as a capping agent with which to encapsulate QRC. An inclusion of QRC into the cavity of β-CD is analyzed by molecular docking studies. The QRC encapsulation by β-CD-K MOFs is analyzed using various methods, such as Fourier transform infrared spectroscopy (FT-IR), nuclear magnetic resonance spectroscopy (NMR), scanning electron microscopy (SEM), X-ray diffraction (XRD), X-ray photoelectron spectroscopy (XPS), and differential scanning calorimetry (DSC). The objective of the study is to explore the potential of β-CD-K MOFs in order to encapsulate and stabilize QRC in the application of biomedicine applications.

## 2. Results and Discussion

### 2.1. Interaction of QRC:β-CD in the Virtual State

Prior to studying the QRC:β-CD-K MOFs, it is necessary to analyze the interaction of the QRC with the β-CD, the energy parameter in detail, and the hydrogen bonding interaction.

#### 2.1.1. Single-Point Energy Computation for the Energetically Favorable Model

The orientation of the QRC toward the host for inclusion complexes (ICs) in the virtual state is determined using single-point energy calculations [33]. For a QRC, there are two different kinds of orientation, represented by orientation I and orientation II. PM3 is used to calculate the single point energy for orientation I—which belongs to the orientation of the benzene ring part of the QRC towards the 2° rim of the β-CD, denoted as ICs of orientation I (Figure 1A,C)—and orientation II, which belongs to the orientation of the quinone ring part of the QRC towards the 2° rim of the β-CD, denoted as ICs of orientation II (Figure 1B,D).

Using the following equation, a semi-empirical technique is used to determine the complexation energy (ΔE) between QRC and β-CD for the optimum energy structure.
ΔE = E_QRC:β-CD_ − (E_β-CD_ + E_QRC_)

The energies of free and optimized β-CD, QRC, and QRC:β-CD are denoted as E_β-CD_, E_QRC_, and E_QRC:β-CD_, respectively. According to Table 1, negative complexation energy shows that the inclusion process is thermodynamically advantageous in a vacuum. Orientations I and II have complexation energies of −52.249 and −33.870 kJ/mol, respectively. There is a thermodynamically advantageous and also more stabilized structure, based on the complexation energy (ΔE) of ICs, with orientation I than orientation II. Figure 2 depicts the hydrogen bonding interactions for both orientations with various atoms, and Table 2 lists the total results.

#### 2.1.2. E_HOMO_-E_LUMO_ of Frontier Molecular Orbitals of ICs

It is possible to assess the stability of the ICs using the highest occupied molecular orbital (HOMO) and lowest unoccupied molecular orbital (LUMO) [34]. Understanding the stability of the molecules depends on the energy difference between the HOMO and LUMO (EHOMO-ELUMO). The ICs with orientation II are a highly favorable and stabilized model because the value of EHOMO-ELUMO is higher for the ICs than for the ICs with orientation I (Table 3 and Figure 3). In general, ICs with high EHOMO-ELUMO values are more stable [35,36].

### 2.2. Spectral Analysis of ICs Based MOFs

#### 2.2.1. FT-IR Spectral Analysis

The FT-IR spectrum of β-CD is consistent with the literature [37,38] as are the predominant peaks at around 3424 cm^−1^ (caused by the symmetric stretching of the hydroxyl groups), 2929 cm^−1^ (caused by the asymmetric stretching of the CH_2_ band), 1640 cm^−1^ (caused by the presence of water molecules in the CD cavity), 1430, 1369, and 1155 cm^−1^ (caused by the deformation vibrations of the C–H bonds in the primary and secondary hydroxyl groups of β-CD), and 1031 cm-1 (caused by the C–C stretching vibrations). These peaks can all be observed in the FT-IR spectrum of β-CD-K MOFs, with slightly shifted positions, which confirmed the presence of β-CD in the formed MOFs. Additionally, the shift in the bands indicated the interaction between β-CD and K ions. Consequently, the parent β-CD skeleton structure is maintained in β-CD-K MOFs, as evidenced by the comparison of the FT-IR spectra of β-CD and β-CD-K MOFs (Figure 4). The FT-IR spectrum of QRC agreed with the literature [39,40]. The IR signal appears to be overlapped with β-CD for MOFs lacking QRC. As a result, after forming MOFs with K ions, there was no shift and no peak disappearance [38]. However, the MOFs with QRC showed a little variation in the stretching frequencies. It is interesting to note that the stretching peak for the β-CD developed at 1645 cm^−1^, and its MOFs were mostly caused by the presence of water molecules in the interior cavity of the β-CD. Moreover, the MOFs maintained their ability of cavity to hold water molecules. When MOFs contain a QRC, the peak is significantly diminished, and it is thought that the QRC creates ICs by replacing the water molecules (shoulder). In the spectrum of QRC:β-CD-K MOFs, certain distinctive QRC bands can be observed, suggesting that the QRC has been loaded into the β-CD-K MOFs (Appendix A). Unfortunately, due to the low weight percentage of QRC in the MOFs, several peaks cannot be clearly distinguished. The modifications also validate the successful packing of QRC by β-CD-K MOFs.

#### 2.2.2. NMR Spectral Analysis

Figure 5 shows the ^1^H NMR spectra of β-CD-K and QRC:β-CD-K MOFs. A signal at 8.31 ppm as a singlet is observed in the β-CD (Figure 5A), and is attributed to the hydroxy proton linked to the methylene group. The methine proton of the carbon atom connected to the hydroxy methylene group is responsible for the triplet which is observed at 4.45 ppm, with a coupling constant value of about 12 Hz. Further methine and methylene protons of the β-CD ring are responsible for the three multiplets observed at 3.63, 3.56, and 3.30 ppm. Equatorial hydroxy protons are responsible for the doublets with modest coupling constants at 5.67 and 4.82 ppm in the β-CD ring. It is simple to attribute the residual signal, which has a coupling constant of 5.73 ppm and 6.6 Hz, to the equatorial hydrogen of the β-CD.

The presence of aromatic protons in the MOFs in the range of 6.00–7.50 ppm provides proof that the organic molecule quercetin is present in the MOFs (Figure 5B). The substituent and electronegative effects of the groups present in the molecule are used to allocate each of these protons individually. The aromatic protons appear as doublets at 6.18, 6.40, and 6.88 in the NMR spectrum of the free QRC because they are present between the oxygen moiety (Appendix A). The two neighboring protons in the phenyl group yield a doublet at 7.54 and a doublet at 7.67 ppm. At 9.37 and 12.48 ppm [41], the hydroxy protons appear as the wide signal. Three doublets for four protons are found in the MOFs at 7.51, 7.22, and 6.68 ppm, as well as a broad singlet at 6.19 ppm. The aromatic protons of the QRC moiety of the MOFs are responsible for these proton signals. One of the aromatic ring’s hydroxy protons yields a wide singlet at 12.34 ppm. The other hydroxy protons are at the shielding region at 5.91, 5.75, and 5.32 ppm; however, they also appear as broad signals with low intensity, indicating that the QRC is very low in β-CD-K MOFs (Appendix A). Similarly, the neighboring proton in the phenyl group of QRC is somewhat shifted and yields the doublets of a doublet at 7.56 ppm. This shifting may be the result of the interaction between the QRC and the β-CD-K MOFs.

#### 2.2.3. XRD Analysis

The high crystallinity of the β-CD-K MOFs is revealed by the XRD results in Figure 6 [11]. β-CD-K MOFs have demonstrated three main and predominant peaks at 8.9°, 12.88°, and 18.8° [42,43,44]. The diffraction patterns obtained from the experiment are in excellent agreement with the simulated diffraction peaks, proving that the compound is in pure phases and suggesting that the different orientations of the sample may be the cause of the disparity in peak intensities between the simulated and experimental patterns. The three specific peaks for the QRC appeared at 12.56°, 16.2°, and 27.11°, denoting that the QRC is crystalline in nature. The predominant peaks reappeared in the QRC:β-CD-K MOFs, and the peak at 27.0° is still visible, but with decreased strength. The absence of the peaks associated with the crystalline QRC-loaded ICs suggests that the QRC is integrated with an amorphous state.

#### 2.2.4. FE-SEM Image Analysis

By FE-SEM image analysis, the morphologies of the formed MOFs are clearly visualized [45,46] (Figure 7). As per the visualization of the SEM images, MOFs revealed a powdered substance without cubic morphologies. Images of MOFs without QRC loading suggested that the smaller, irregularly formed crystals may be a result of the manufacturing process for MOFs. The tiny crystalline appearance of the QRC is entirely destroyed during the loading phase, and is replaced with a powdered appearance. The grinding process is primed to cause the majority of the particles in the MOFs to lose their form. In order to learn more about the elements present in the generated MOFs, elemental mapping and EDX are used (Figure 8). Figure 8A,F show that the MOFs contain the elements C, O, and K. Additionally, the ratios of the various components are supplied. In the presence or absence of QRC, there are no appreciable variations in the atomic weight percentages, particularly for C and O. The elemental analysis of individual C, O, and K atoms is also provided, and confirmed their presence in the formed MOFs using elemental mapping.

#### 2.2.5. XPS Analysis

By comparing QRC:β-CD-K MOFs samples, the assumed binding between CD-MOFs and the QRC is clarified (Table 4 and Figure 9). According to the findings, with the support of XPS measurements, CD and the MOFs are bonded together by interactions because O 1s, K 2p, and C 1s of the MOFs are observed in the survey spectrum of β-CD-K MOFs. The binding energy for the C 1s peaks changed from 285.68 eV for the β-CD-K MOFs to 285.88 eV for the QRC:β-CD-K MOFs. Moreover, the composition of QRC:β-CD-K MOFs increased from 54.90% to 55.63%. The peaks focused at 284 and 286 eV, which correspond to C–C and C–O bonds, respectively, could be the result of unreacted CDs on QRC [47]. With no discernible changes in the atomic ratios, the fitted O 1s peak’s binding energy (at 532.08 eV for β-CD-K MOFs samples) rose. A drop in the electron density surrounding the O element, and an increase in the binding energy of O-K, comes from the QRC:β-CD-K MOFs’ 1s atomic ratio becoming larger than that of K 2p as growing time increases [48,49]. Our findings showed that, as the growing time lengthened, more β-CD-K MOFs formed on the surface of QRC treated with O_2_ plasma by binding with the K element. Peaking at 292.3 eV, K 2p_3/2_ is indicated. In the high-resolution spectra of K 2p, the peaks at 292.3 and 295.6 eV, which are separated by 3.3 eV, are assigned to K 2p_3/2_ and K 2p_1/2_ of K^+^, respectively [47]. Since the hydroxyl groups on the β-CD-K MOFs linked or coupled with K by extra cross-linking of β-CD to produce dendricolloids, due to a prolonged growth period, the 80 h QRC:β-CD-K MOFs showed the highest K 2p of 2.04% atomic ratio among QRC:β-CD-K MOFs [50]. This corroborated the findings of the FT-IR analysis [51] and suggested that β-CD served as a seed in the formation of β-CD-K MOFs [52]. However, after the adsorption of QRC, their binding energies and relative contents (Table 4) considerably changed, suggesting that interactions took place during the adsorption of QRC.

#### 2.2.6. DSC Analysis

Figure 10 displays the DSC curves for both MOFs in the temperature range of 40.0 to 250 °C. Here, the β-CD and the raw QRC are not visible. Two endothermic peaks in the range of 25–500 °C are visible on the DSC curve of the QRC [50]. The initial peak, which is connected to the discharge of water from QRC, is recorded in the 109–136 °C temperature range [53]. As this temperature is substantially higher than the boiling point of water, QRC’s hydrogen bonds with the water molecules must be quite strong [54]. The melting point of QRC is revealed by the second endothermic event, which occurred between 318 and 328 °C [55]. A deterioration process that began at about 300 °C is followed by a very broad endothermic peak in the β-CD spectrum between 48 °C and 155 °C [56]. It is discovered that the DSC of β-CD-K MOFs had an endothermic peak at about 204.57 °C and 162.4 J/g of enthalpy changes at melting (ΔH_melting_), which are likely connected to the interaction of K in the MOFs. There are two endothermic maxima for QRC:β-CD-K MOFs at 177.40 and 216.25 °C, with the values of ΔH_melting_ (blue symbols covered in the figure) found to be 8.659 and 38.31 J/g, respectively. Moreover, neither of these two temperatures corresponds to the melting points of QRC. This might be brought on by MOF deterioration.

### 2.3. Antibiofilm Potency and SEM Analysis of QRC:β-CD-K MOFs Treated C. albicans

*C. albicans* was used in a biofilm assay to examine the antibiofilm potency of QRC:β-CD-K MOFs. Figure 11 displays a dose-dependent antibiofilm inhibition following the application of QRC:β-CD-K MOFs at dosages of 1, 5, 25, and 50 µg/mL. After 24 h of incubation, QRC:β-CD-K MOFs prevented the production of biofilms at a concentration of 25 µg/mL, inhibiting >54 ± 6.9% of it. Additionally, when the dose of QRC:β-CD-K MOFs was increased to 50 µg/mL, significant *C. albicans* biofilm suppression was observed (>82 ± 2.7%) with no impact on cell proliferation, whereas QRC and β-CD prevented the production of biofilms 24 ± 1.9 and 8 ± 0.8%, respectively, at a concentration of 50 µg/mL (Appendix A). It was discovered that the MICs of QRC:β-CD-K MOFs against *C. albicans* are 200 µg/mL. *C. albicans* is susceptible to QRC:β-CD-K MOFs, which was confirmed by a well diffusion assay. QRC:β-CD-K MOFs showed a clear zone for the activity against *C. albicans.* The diameters of inhibition zones after the treatment are consolidated in Appendix A, and representative images are presented in Appendix A. For example, in *C. albicans* cells, the zones of inhibition were determined to be 22.0 ± 1.1 mm and 28.0 ± 1.2 mm against QRC:β-CD-K MOFs at 500 and 1000 µg/mL, respectively. The antifungal efficacies of QRC:β-CD-K MOFs, QRC, and β-CD were further confirmed by determining the percentage of cell survival at different concentrations (Appendix A). The results revealed that the % of cell survival of *C. albicans* cells decreases with an increase in the concentration of QRC and QRC:β-CD-K MOFs. Particularly, MFC (Minimum Fungicidal Concentration) of QRC:β-CD-K MOFs against *C. albicans* is 400 µg/mL, and is determined using the spread plate method. COMSTAT biofilm analysis (Table 5) and CLSM (Figure 11) both showed biofilm reductions. For example, compared to untreated controls, QRC:β-CD-K MOFs at 50 µg/mL reduced *C. albicans* biofilm biomasses and mean thicknesses by >91 and 86%, respectively (Table 5). The hyphae transition on the surface of nylon filter sheets was also suppressed by QRC:β-CD-K MOFs in PDB medium at 25 and 50 µg/mL, as presented in Figure 12. *C. albicans* was present in the untreated control biofilm group, along with sizable hyphal cells. The QRC:β-CD-K MOFs-treated cells, in contrast, are more than just yeast cells with uncommon hyphae/pseudohyphae. Additionally, the results of the cell aggregation and hyphal growth are consistent with the antibiofilm activity that was found in the treatment group. Therefore, QRC:β-CD-K MOFs had no effect on the growth of planktonic cells, while inhibiting the development of hyphae and biofilm in *C. albicans*.

## 3. Materials and Methods

### 3.1. Materials

β-Cyclodextrin (β-CD), potassium hydroxide (KOH), quercetin (QRC) (>99.5% purity), methanol (MeOH), and dichloromethane (CH_2_Cl_2_) of analytical grade were purchased from Sigma-Aldrich Co. Ltd., Seoul, Republic of Korea. Deionized water was prepared in our laboratory.

### 3.2. Preparation of β-CD-K MOFs

The synthesis of β-CD-K MOFs was carried out using an improved methanol vapor diffusion method with 493 mg of β-CD (in 10 mL of deionized water), and 182 mg of KOH (in 10 mL of deionized water) in an aqueous solution (Figure 1). The solution was then stirred for 30 min, and filtered through a 0.45 m membrane into a glass vessel containing 0.5 mL methanol, which was sealed in a beaker with MeOH. The entire solution was then kept undisturbed for 10 days, while the surrounding 40 mL of methanol facilitated the methanol diffusion process. The white block crystals were then produced, washed three times with isopropanol, and activated by solvent exchange, which was accomplished by soaking β-CD-K MOFs in dichloromethane (CH_2_Cl_2_) for three days. The sample was then vacuum dried at 40 °C overnight. Thereafter, the final products were named β-CD-K MOFs.

### 3.3. QRC Adsorption Process (QRC:β-CD-K MOFs)

It was investigated how QRC adsorbed on β-CD-K MOFs using the impregnation process. A heterogeneous solution was produced by impregnating 30 mg of MOFs and 2 mg of QRC in 10 mL of ethanol solution for 48 h, during which time the solution was shaken at 400 rpm. Centrifugation was used to separate the QRC-loaded ICs, after which the solid was repeatedly rinsed with ethanol and left to dry at 40 °C overnight in a vacuum. Thereafter, the final products were named QRC:β-CD-K MOFs.

### 3.4. Quantum Mechanical Calculations

PM3 in Gaussian 16 was used to perform the semi-empirical quantum mechanical computations.

### 3.5. Experimental Section and Materials for Biofilm

The materials required for the biofilm experiment and the procedure for the biofilm formation are provided in the Appendix A.

### 3.6. Antibiofilm Potency of QRC:β-CD-K MOF against C. albicans

With the help of the crystal violet staining procedure, a biofilm assay was conducted [57]. In a nutshell, a *C. albicans* DAY185-injected overnight culture at 37 °C with shaking was employed. From each overnight culture, *C. albicans* was injected into PDB at a 1:25 dilution ratio. The QRC:β-CD-K MOF was applied to PDB-infected cultures at a concentration of (0–50 μg/mL). The microtiter plates were incubated for 24 h at 37 °C. The formation of the biofilm was then confirmed by staining the microtiter plates with 0.1% crystal violet for 30 min, washing them periodically with distilled water, and then adding 300 μL of 95% ethanol to each well. Each 96-well microtiter plate’s absorbance was measured at 570 nm using an Elisa microplate reader from Biobase (Jinan, China).

### 3.7. Architecture of C. albicans Biofilm

The Phenotypic differences and biofilm architecture of *C. albicans* DAY185 on LCS [58,59,60] are provided in the Appendix A.

### 3.8. Biofilm Observations by Confocal Laser Scanning Microscopy

For the CLSM assay, single-strain biofilms of *C. albicans* were produced in 96-well plates (with or without QRC:β-CD-K MOFs) at 37 °C for 24 h without shaking [58,59]. Detailed information is provided in the Appendix A.

### 3.9. Characterization Techniques

The formed MOFs were characterized by analytical techniques, such as FT-IR, proton NMR, XRD, FE-SEM, XPS, and DSC. The complete instrumental details are provided in the Appendix A.

## 4. Conclusions

An enhanced methanol vapor diffusion technique was used to create β-CD-K MOFs. In order to produce a material with ICs as effective as those that restricted *C. albicans* hyphae growth and biofilm formation while having no effect on planktonic cell growth, the antibacterial plant flavonoid QRC was loaded onto β-CD-K MOFs. Moreover, QRC:β-CD-K MOFs inhibited >54.6% of the development of biofilms at a concentration of 25 g/mL. This finding opens up new possibilities for the creation of supramolecular β-CD-based MOFs for usage in biological investigation, and also in drug development.

## Data Availability

The datasets used or analyzed during the current study are available from the corresponding author upon reasonable request.

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
