# Peer review of "Supramolecular β-Cyclodextrin-Quercetin Based Metal–Organic Frameworks as an Efficient Antibiofilm and Antifungal Agent"

_molecules, 2023, doi:10.3390/molecules28093667_

Round 1

Reviewer 1 Report

Quercetin is a natural flavonoid known for multi pharmacological effects. However, its limited physicochemical properties challenge its druggability and direct human use. Therefore, there is research interest in developing strategies to improve its physicochemical properties. The current paper discusses a strategy where quercetin is loaded into cyclodextrin molecular frame and tested for antibiofilm and antifungal activities. There are some points that the authors need to address.

1. Typographical errors or choice of words.

For example,

Page 2, line 60, "Fortunately. By forming inclusion complexes (ICs) with both native."

Page 1, line 10, "The loading of pharmaceutically important material has recently been done using metal-", the "pharmaceutically important material" should be a "drug or medicinally active compound"

2. Labelling of peak characterization should be done in FTIR spectra in Figure 4. Or at least show them individually as sub-figures, a, b, c, d.

3.  In Figure 5, there seems to be a problem in NMR signals integration; please check again.

4. Were there supposed to be spectral signals for quercetin in Figure 5(B)? Please justify.

5. There seems to be a clear visibility issue with Figure 7 images. Please recheck them, and provide them with a better resolution.

6. There is little to no information provided by the author about the loading of quercetin into cyclodextrin. Please add more context related to the interfacial interaction of quercetin with cyclodextrin in support of the spectral data reported in this manuscript. 

Author Response

We are highly thankful to the Editor and the Reviewers for the insightful and constructive comments on our manuscript (Manuscript ID:molecules-2319177) entitled “Supramolecular β-cyclodextrin-quercetin based metal-organic frameworks as an efficient antibiofilm and antifungal agent”. As suggested by the Editor and the Reviewers, we have addressed the issues raised and modified the manuscript accordingly.

Dear Ms. Katarina Modic,

Assistant Editor.

We appreciate your kind consideration and constructive comments on our manuscript. As suggested by the editor and reviewers, we have markedly modified the manuscript. The editor’s and reviewer’s questions have been repeated in black text and our response follows in blue. In the revised manuscript, changes are in red.

Comments from the reviewers

Reviewer 1

Quercetin is a natural flavonoid known for multi pharmacological effects. However, its limited physicochemical properties challenge its druggability and direct human use. Therefore, there is research interest in developing strategies to improve its physicochemical properties. The current paper discusses a strategy where quercetin is loaded into cyclodextrin molecular frame and tested for antibiofilm and antifungal activities. There are some points that the authors need to address.

We would like to thank the Reviewers for their thorough reading of this manuscript and for the recommendations that have helped us improve the manuscript’s quality and scientific value.

  1. Typographical errors or choice of words.

For example,

Page 2, line 60, "Fortunately. By forming inclusion complexes (ICs) with both native."

Page 1, line 10, "The loading of pharmaceutically important material has recently been done using metal-", the "pharmaceutically important material" should be a "drug or medicinally active compound"

Author responses : Typographical errors in the whole manuscript are now corrected.

  1. Labelling of peak characterization should be done in FTIR spectra in Figure 4. Or at least show them individually as sub-figures, a, b, c, d.

Author responses : As per the comment, the selective and predominant frequencies are only labeled in the spectra. However, all the frequencies (peaks) are mentioned in the individual spectra and provided in Figure S1.

  1. In Figure 5, there seems to be a problem in NMR signals integration; please check again.

Author responses : NMR signals are now rechecked and now corrected in the revised manuscript.

  1. Were there supposed to be spectral signals for quercetin in Figure 5(B)? Please justify.

Author responses : Spectral signals for quercetin are very low, as it is loaded in a small quantity. For better visibility, we raised spectra and provided them in the supplementary documents, Figure S3. 

  1. There seems to be a clear visibility issue with Figure 7 images. Please recheck them, and provide them with a better resolution.

Author responses : Now the visibility of Figure 7 is improved.

  1. There is little to no information provided by the author about the loading of quercetin into cyclodextrin. Please add more context related to the interfacial interaction of quercetin with cyclodextrin in support of the spectral data reported in this manuscript. 

Author responses : To understand how QRC and β-CD interact, virtual proof is offered. The manuscript includes a thorough discussion. Additionally, the published article has already covered the experimental evidence for the inclusion interaction of QRC with β-CD. As a result, we concentrated on the synthesis, description, and biological utility of the MOFs.

Reviewer 2 Report

1. Introduction is short

2. In the FTIR spectrum, add the frequencies inside the images

3. Include the NMR result for free quercetin also

4. Include the anticandidal plate images and also MIC confirmation by resazurin assay

5. Include the bioflim data for quercetin and β-CD

6. Check the SI units throughout the manuscript

7. Include more recent reference

8. Discuss the results with more relevant references

Author Response

We are highly thankful to the Editor and the Reviewers for the insightful and constructive comments on our manuscript (Manuscript ID:molecules-2319177) entitled “Supramolecular β-cyclodextrin-quercetin based metal-organic frameworks as an efficient antibiofilm and antifungal agent”. As suggested by the Editor and the Reviewers, we have addressed the issues raised and modified the manuscript accordingly.

Dear Ms. Katarina Modic,

Assistant Editor.

We appreciate your kind consideration and constructive comments on our manuscript. As suggested by the editor and reviewers, we have markedly modified the manuscript. The editor’s and reviewer’s questions have been repeated in black text and our response follows in blue. In the revised manuscript, changes are in red.

Comments from the reviewers

Reviewer 2

We would like to thank the Reviewers for their thorough reading of this manuscript and for the recommendations that have helped us improve the manuscript’s quality and scientific value.

  1. Introduction is short

Author responses : The introduction is revised with the scope and need of the present study.

  1. In the FTIR spectrum, add the frequencies inside the images

Author responses : As per the comment, the selective and predominant frequencies are only labeled in the spectra. However, all the frequencies (peaks) are mentioned in the individual spectra and provided in Figure S1.

  1. Include the NMR result for free quercetin also

Author responses : NMR spectrum for the free QRC is provided in the supplementary documents (Figure S2).

  1. Include the anticandidal plate images and also MIC confirmation by resazurin assay

Author responses : We are thankful to the reviewer for their insightful comments and suggestions. Due to some technical issues, we were unable to perform MIC confirmation by resazurin assay experiment. But to fulfill the reviewer’s comment and suggestion we calculated % cell survival in the biofilm inhibition experiment using the spread plate technique and dose-dependent anticandidal activity of QRC:β-CD-K MOFs was summarized in a revised manuscript.

  1. Include the bioflim data for quercetin and β-CD

Author responses : As suggested, we have included biofilm data for quercetin and β-CD in supplementary section as figure S7.

  1. Check the SI units throughout the manuscript

Author responses : SI units are now checked and corrected in the revised manuscript.

  1. Include more recent reference

Author responses : More recent references are now included in the revised manuscript.

  1. Discuss the results with more relevant references

Author responses : Relevant references are now included in the discussion in the revised manuscript.

Reviewer 3 Report

In this work quercetin (QRC) is loaded onto β-Cyclodextrin Metal-Organic Frameworks (β-CD-K MOFs) to create the composite containing inclusion complexes (ICs). QRC:β-CD-K MOFs samples are characterized using various spectroscopy and microscopy methods. The Candida albicans DAY185 strain is used to evaluate the anti-biofilm of QRC:β-CD-K MOFs and the antifungal effectiveness of the prepared samples. This study suggested that supramolecular β-CD-based MOFs could be useful antifungal agent in various biological and pharmaceutical applications.

 The results presented in this study are interesting; however, the authors need to consider the comments listed below before re-submission of the paper.

Major issues:

The English is poor and there are many syntax errors. Some sentences are incomplete and the manuscript text is not clearly written. In particular, the abstract and introduction sections are poorly written. Moreover, first and last paragraphs need to be accordingly revised. In addition, the authors need to emphasise novelty in this study and they need clearly to explain the aim of this work.

Minor issue:

 The major FTIR peaks in all samples need to be labelled in Figure 4.

Author Response

We are highly thankful to the Editor and the Reviewers for the insightful and constructive comments on our manuscript (Manuscript ID:molecules-2319177) entitled “Supramolecular β-cyclodextrin-quercetin based metal-organic frameworks as an efficient antibiofilm and antifungal agent”. As suggested by the Editor and the Reviewers, we have addressed the issues raised and modified the manuscript accordingly.

Dear Ms. Katarina Modic,

Assistant Editor.

We appreciate your kind consideration and constructive comments on our manuscript. As suggested by the editor and reviewers, we have markedly modified the manuscript. The editor’s and reviewer’s questions have been repeated in black text and our response follows in blue. In the revised manuscript, changes are in red.

Comments from the reviewers

Reviewer 3

In this work quercetin (QRC) is loaded onto β-Cyclodextrin Metal-Organic Frameworks (β-CD-K MOFs) to create the composite containing inclusion complexes (ICs). QRC:β-CD-K MOFs samples are characterized using various spectroscopy and microscopy methods. The Candida albicans DAY185 strain is used to evaluate the anti-biofilm of QRC:β-CD-K MOFs and the antifungal effectiveness of the prepared samples. This study suggested that supramolecular β-CD-based MOFs could be useful antifungal agent in various biological and pharmaceutical applications.

The results presented in this study are interesting; however, the authors need to consider the comments listed below before re-submission of the paper.

We would like to thank the Reviewers for their thorough reading of this manuscript and for the recommendations that have helped us improve the manuscript’s quality and scientific value.

Major issues:

The English is poor and there are many syntax errors. Some sentences are incomplete and the manuscript text is not clearly written. In particular, the abstract and introduction sections are poorly written. Moreover, first and last paragraphs need to be accordingly revised. In addition, the authors need to emphasise novelty in this study and they need clearly to explain the aim of this work.

Author responses : English in the whole manuscript is now revised and also syntax errors are also reduced. The abstract and Introduction part is now rewritten. The novelty and aim of this work are also now revised.

Minor issue:

The major FTIR peaks in all samples need to be labelled in Figure 4.

Author responses : As per the comment, the selective and predominant frequencies are only labeled in the images. However, all the frequencies are labeled in the individual spectra and provided in Figure S1.

Round 2

Reviewer 2 Report

-

Reviewer 3 Report

The authors revised the manuscript as per reviewer’s comments. The revised manuscript is acceptable for publication.